# Computational simulations and Ca²⁺ imaging reveal that slow synaptic depolarizations (slow EPSPs) inhibit fast EPSP evoked action potentials for most of their time course in enteric neurons

**Parvin Zarei Eskikand**[¤a], **Katerina Koussoulas**[¤b], **Rachel M. Gwynne**[¤b], **Joel C. Bornstein**[¤b]*

Department of Physiology, School of Biomedical Sciences, University of Melbourne, Parkville, Australia

¤a Current address: Department of Biomedical Engineering, Faculty of Engineering and Information Technology, University of Melbourne, Parkville, Australia
¤b Current address: Department of Anatomy and Physiology, School of Biomedical Sciences, University of Melbourne, Parkville, Australia
* j.bornstein@unimelb.edu.au

## Abstract

Transmission between neurons in the extensive enteric neural networks of the gut involves synaptic potentials with vastly different time courses and underlying conductances. Most enteric neurons exhibit fast excitatory post-synaptic potentials (EPSPs) lasting 20–50 ms, but many also exhibit slow EPSPs that last up to 100 s. When large enough, slow EPSPs excite action potentials at the start of the slow depolarization, but how they affect action potentials evoked by fast EPSPs is unknown. Furthermore, two other sources of synaptic depolarization probably occur in enteric circuits, activated via GABA_A or GABA_C receptors; how these interact with other synaptic depolarizations is also unclear. We built a compartmental model of enteric neurons incorporating realistic voltage-dependent ion channels, then simulated fast EPSPs, slow EPSPs and GABA_A or GABA_C ligand-gated Cl⁻ channels to explore these interactions. Model predictions were tested by imaging Ca²⁺ transients in myenteric neurons ex vivo as an indicator of their activity during synaptic interactions. The model could mimic firing of myenteric neurons in mouse colon evoked by depolarizing current during intracellular recording and the fast and slow EPSPs in these neurons. Subthreshold fast EPSPs evoked spikes during the rising phase of a slow EPSP, but suprathreshold fast EPSPs could not evoke spikes later in a slow EPSP. This predicted inhibition was confirmed by Ca²⁺ imaging in which stimuli that evoke slow EPSPs suppressed activity evoked by fast EPSPs in many myenteric neurons. The model also predicted that synchronous activation of GABA_A receptors and fast EPSPs potentiated firing evoked by the latter, while synchronous activation of GABA_C receptors with fast EPSPs, potentiated firing and then suppressed it. The results reveal that so-called slow EPSPs have a biphasic effect being likely to

**Data Availability Statement:** Codes for the models used in the simulations can be obtained from https://github.com/ParvinZE/Slow-EPSPs.git and the raw data used for the intracellular recording Figures and the Ca imaging analysis can be obtained from https://melbourne.figshare.com/articles/dataset/Role_of_slow_EPSPs_in_regulating_firing_triggered_by_fast_EPSPs_in_enteric_neurons_-_Ca_imaging_and_intrracellular_recording_raw_data/19784584 Dataset. https://doi.org/10.26188/6287433ec0145.

**Funding:** Funding provided by National Institutes of Health SPARC (Stimulating Peripheral Activity to Relieve Conditions) program, OT2OD023859 (JCB) https://ncats.nih.gov/sparc; and National Health Medical Research Council (Australia), project GNT1158952 (JCB) https://www.nhmrc.gov.au/. The funders had no role in study design, data collection and analysis, decision to publish, or preparation of the manuscript.

**Competing interests:** The authors have declared that no competing interests exist.

suppress fast EPSP evoked firing over very long periods, perhaps accounting for prolonged quiescent periods seen in enteric motor patterns.

## Author summary

The gastrointestinal tract is the only organ with an extensive semi-autonomous nervous system that generates complex contraction patterns independently. Communication between neurons in this "enteric" nervous system is via depolarizing synaptic events with dramatically different time courses including fast synaptic potentials lasting around 20–50 ms and slow depolarizing synaptic potentials lasting for 10–120 s. Most neurons have both. We explored how slow synaptic depolarizations affect generation of action potentials by fast synaptic potentials using computational simulation of small networks of neurons implemented as compartmental models with realistic membrane ion channels. We found that slow synaptic depolarizations have biphasic effects; they initially make fast synaptic potentials more likely to trigger action potentials, but then actually prevent action potential generation by fast synaptic potentials with the inhibition lasting several 10s of seconds. We confirmed the inhibitory effects of the slow synaptic depolarizations using live $Ca^{2+}$ imaging of enteric neurons from mouse colon in isolated tissue. Our results identify a novel form of synaptic inhibition in the enteric nervous system of the gut, which may account for the vastly differing time courses between signalling in individual gut neurons and rhythmic contractile patterns that often repeat at more than 60 s intervals.

## Introduction

The enteric nervous system (ENS) is the extensive network of interconnected neurons contained entirely within the wall of the gastrointestinal tract (GIT). The ENS interacts with the central nervous system but can operate semi-autonomously [1] and, even though it is modulated by extrinsic input, it does not necessarily depend on this input for its complex functions. Therefore, the ENS requires a complete circuit of intrinsic neurons to function and control responses to stimuli such as stretch, distortion of the mucosa and nutrients and other chemicals in the lumen.

The ENS contains at least 20 distinct classes of neurons and several classes of glial cells depending on the classification system used [2,3]. The main categories include populations of intrinsic sensory neurons (ISNs), orally directed ascending interneurons, anally directed descending interneurons, excitatory and inhibitory motor neurons supplying the circular muscle, separate groups of excitatory and inhibitory motor neurons innervating the longitudinal muscle and secretomotor neurons. In the small and large intestines, the cell bodies of these neurons are in the myenteric plexus and the submucosal plexus [4,5]. It is widely accepted that the ISNs (also known as intrinsic primary afferent neurons, IPANs) have a common electrophysiological signature in guinea-pigs and rodents [6]. But it has not been possible to identify distinct electrophysiological signatures for the other subtypes, which are collectively termed S neurons (for a historical review see [7]). These neurons exhibit a variety of distinct synaptic potentials, which are depolarizing in most cases (for review see [8] There are four different types of synaptic potential identified in the ENS based on their temporal dynamics and their depolarizing or hyperpolarizing effect on the membrane potential. These include three

classes of depolarizing potential: fast excitatory synaptic potentials (EPSPs) with time courses of 30–50 ms intermediate EPSPs, which can last 150 ms to 2.5 s; and slow synaptic depolarizations, usually termed slow EPSPs, which last 10 s to minutes. There are also hyperpolarizing inhibitory postsynaptic potentials (IPSPs) with several subtypes based on their pharmacology and underlying conductance changes [8–10].

Fast EPSPs have been observed in at least 60% of myenteric and 90% of submucosal neurons. These are analogous to EPSPs mediated by glutamate in the central nervous system (CNS), but in the ENS the fast EPSPs are predominantly mediated by acetylcholine (ACh) acting on nicotinic acetylcholine receptors (nAChR) [8,11–15]. Other transmitters for which there is strong electrophysiological evidence are ATP (or a related purine) acting via P2X receptors (P2XR) and serotonin (5-hydroxytryptamine, 5-HT) acting via 5-HT$_3$ receptors. [14] Activation of nAChR, P2XR and 5-HT$_3$R all produce increases in cation conductance with reversal potentials close to 0 mV [8,14]. Fast EPSPs mediated by ATP acting on P2X receptors and 5-HT acting on 5-HT$_3$ receptors have also been observed in enteric neurons [16–18]. The other prominent depolarizing synaptic events in the ENS are slow EPSPs, which are mediated by various types of metabotropic receptors [8,19–23]. This usually involves closure of K$^+$ channels and consequently a decrease in the K$^+$ conductance [19,22,24]. There is also evidence for prolonged Cl$^-$ conductance increases [25] and indeterminate conductance changes [26,27]. It is generally believed that these slow synaptic depolarizations produce long lasting increases in the excitability of the neurons in which they occur. But there is still no solid proof for this despite it being more than 40 years since slow synaptic depolarizations were first described.

There is also significant evidence that γ-aminobutyric acid (GABA) is a transmitter that contributes to the array of synaptic potentials seen in enteric neurons [28–30]. All three major subtypes of GABA receptors. GABA$_A$, GABA$_B$ and GABA$_C$, are expressed by enteric neurons, and can lead to activation of Ca$^{2+}$ transients in these neurons [30], GABA$_C$ and GABA$_A$ receptors are each pentameric ligand-gated chloride (Cl$^-$) channels (for review [31]). Furthermore, GABA is found in many enteric nerve terminals within the ganglia of various regions of the GI tract, most notably in the colonic myenteric plexus [29] and antagonists of GABA$_A$ receptors can modify spontaneous motor patterns in mouse colon [32]. Nevertheless, the limited electrophysiological studies available have not identified postsynaptic events in mouse colon that can be attributed to GABA acting via its ionotropic receptors.

While GABA$_A$ and GABA$_C$ receptors are both *Cl$^-$* channels, their electrophysiological responses are significantly different [31,33]. GABA$_C$ receptors are very low conductance channels but remain open for a long time as their activation and inactivation kinetics are very slow. The mean opening time of a GABA$_C$ receptor is around 150 ms, eight times slower than GABA$_A$ receptors [33]. The mean opening time for GABA$_A$ receptors is about 25 ms. GABA$_C$ receptors also close eight times slower after the removal of the agonist. In contrast to GABA$_A$ receptors, GABA$_C$ receptors have a very weak desensitization even when the agonist concentration is high [31,34,35]. One aim of our current study was to investigate the interactions between fast EPSPs, slow synaptic depolarizations and synapses acting through GABA$_A$ and GABA$_C$ receptors.

Studies of neuronal responses to activation of slow synaptic depolarizations show that neurons with such inputs fire action potentials during the initial depolarization, but usually fire less or not at all after the initial burst despite the depolarization being sustained [22,26,36]. It is still unknown, however, what this means for the core question for understanding function, that is whether slow synaptic depolarizations make neurons more likely to fire action potentials in response to other forms of synaptic input, notably fast EPSPs. There are two reasons for this. First, there has been a focus on a subset of myenteric neurons, the ISNs, that do not have

prominent fast EPSPs under resting conditions, at least in the small intestine [7,11,37], in contrast to interneurons and motor neurons (S neurons) that do exhibit fast EPSPs. Second, electrical stimulation of nerve trunks within the ENS cannot discriminate between axons that produce slow synaptic depolarization and those that produce fast EPSPs, indeed in some cases the same axon can produce both. Thus, it is impossible to activate one response without also potentially exciting the other.

In this study, we developed a computational model to investigate synaptic interactions between different types of synaptic potentials in enteric interneurons and motor neurons, both of which have prominent fast EPSPs. This computational model predicts that slow synaptic depolarizations inhibit action potentials normally evoked by fast EPSPs mediated by ionotropic receptors. We tested this prediction in *in vitro* experiments using $Ca^{2+}$ imaging as a surrogate measure of neuronal activity. We also investigated whether the slow depolarizations mediated by $GABA_C$ receptors, inotropic receptors that are chloride channels with slow kinetics and a reversal potential 10–15 mV positive to resting membrane potential, can be inhibitory, as suggested by an earlier study from our team [30].

## Results

We developed a biologically realistic model network of neurons to examine the interactions of different types of synapses on the activity profiles of enteric neurons. Individual model neurons were similar in concept to the compartmental model of intrinsic sensory neurons that we had previously built [38]. For details of the individual neuron model and the model channels, see Materials and Methods below.

### Model validation

Conventional intracellular recordings (for methods see [37]) from myenteric neurons in mouse proximal colon reveal that the vast majority of such neurons fire only transiently at the start of an imposed depolarization (Fig 1A), but at up to 150 Hz. Accordingly, we first tested the model by delivering a 500 ms depolarizing current pulse to a single neuron and determined the resultant firing of action potentials when the model incorporated a Kv7.2 potassium channel (Fig 1B) [39] and when it did not (Fig 1C). With the Kv7.2 channel present the firing of the model neuron closely matched the transient firing of the neurons recorded *in vitro*, omission of this channel made the model neuron much more excitable and it fired throughout the imposed depolarization. Thus, for the remainder of this study the Kv7.2 channel was incorporated into the model neurons.

We next identified the responses to activation of individual types of synaptic potential and found a good match between the generic fast EPSP model (Fig 1D) and electrophysiologically recorded fast EPSPs (Fig 1E) and the slow EPSP model (Fig 1G and 1H) and electrophysiologically recorded slow synaptic depolarizations (Fig 1F). A notable feature of the slow synaptic depolarizations in the model is that, when they are large enough to trigger action potentials, the neuron fires only at the start of the depolarization and then remains quiescent despite the continuation of the slow depolarization.

There are no published recordings of synaptic potentials mediated by either $GABA_A$ or $GABA_C$ receptors in enteric neurons, but there is substantial evidence indicating that such potentials exist [28–30,32]. Accordingly, we simulated synaptic potentials mediated by these receptors with their properties set to those given in the Materials and Methods. A key feature here is that the Cl⁻ equilibrium potential in enteric neurons is about -35 mV [40,41], so activation of ionotropic GABA receptors is depolarizing. Modelled $GABA_A$ mediated responses were transient depolarizations, very similar in time course to fast EPSPs (Fig 1I), but

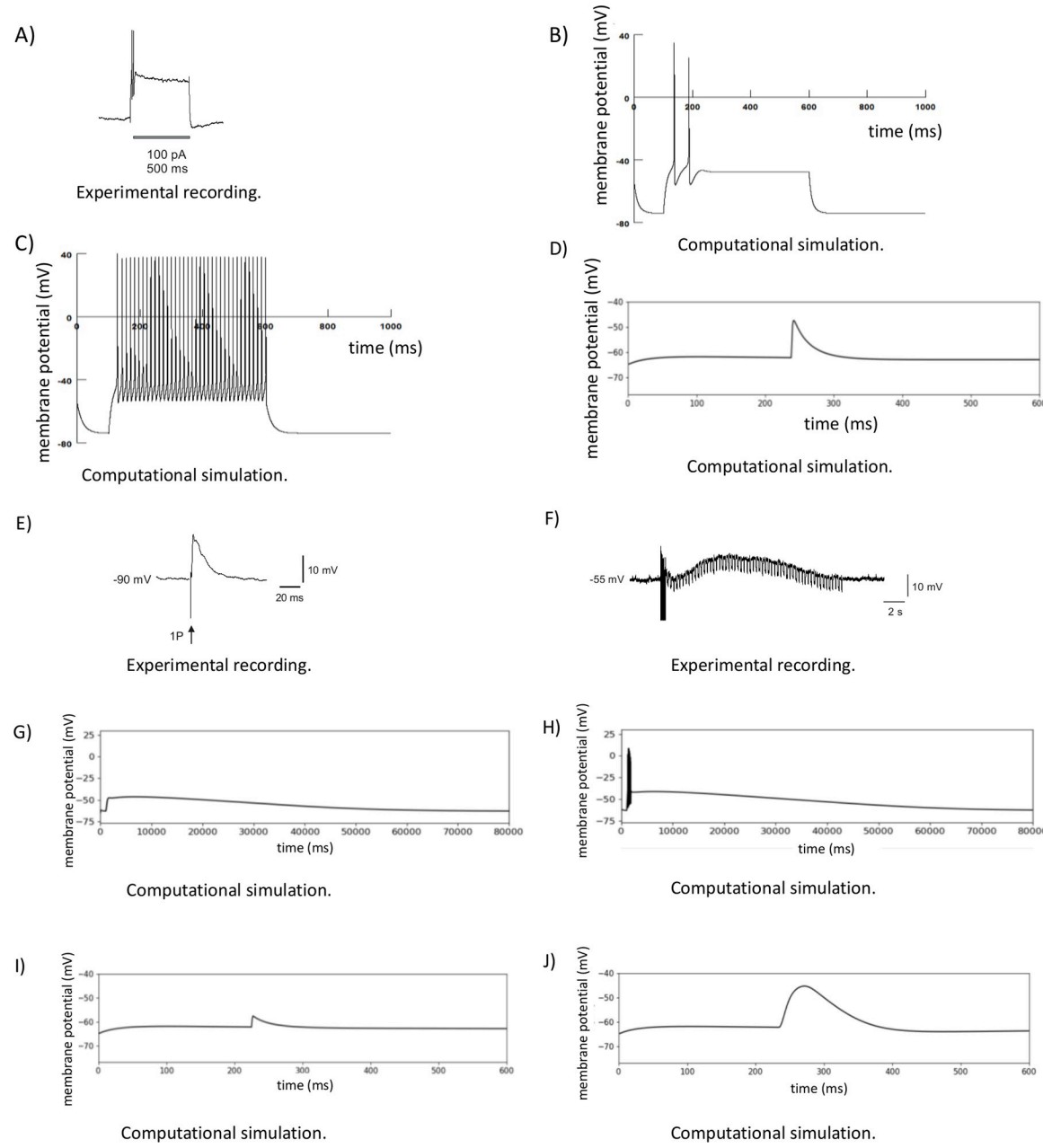

**Fig 1. Validation of the computational model of single enteric neurons.** A) Intracellular recording from myenteric neuron in mouse proximal colon showing the response of this neuron to a 500 ms duration depolarizing current pulse delivered via the recording electrode. The neuron fires only transiently at the start of the depolarization (2 spikes > 120 Hz). B) Response of a modelled neuron to a 500 ms depolarization in the presence of a Kv7.2 channel. C) The response of the same modelled neuron to the 500ms depolarization when the Kv7.2 channel is blocked. D, E) The model fast EPSP and electrophysiologically recorded fast EPSPs, respectively, are virtually indistinguishable. F) An example of electrophysiologically recorded slow EPSP triggered by a train of 10 pulses at 20 Hz, the downward deflections in the membrane potential trace are due to passage of short hyperpolarizing current pulses to monitor input resistance during the slow depolarization. G) A subthreshold slow EPSP generated by the computational model. H) A suprathreshold slow EPSP generated by the computational model. I) The GABA$_A$ mediated response generated by the model. J) The GABA$_C$ mediated response generated by the model. Note, in B, C, D, I and J there is an initial change in membrane potential that represents the model neuron simulation "settling" to its resting membrane potential.

significantly smaller in amplitude for any given synaptic weight due to the more negative reversal potential for the channel (-35 mV vs 0 mV). In contrast, the modelled GABA$_C$ mediated response (Fig 1J) was much slower in time course and, although still limited by the more negative reversal potential, had a larger amplitude at resting membrane potential due to the slower time to peak current.

## Computational modelling of synaptic interactions

To explore the interactions between the different types of depolarizing synaptic potentials, we employed a simple model circuit in which individual neurons using distinct types of transmission converge on a single output neuron (Fig 2A). The input neurons are each driven by a spike generator that activates axons which excite the input neurons to fire action potentials. The sizes of synaptic potentials produced in the recording neuron were adjusted by varying synaptic weights to be either just subthreshold for triggering an output neuron action potential with no other input active or suprathreshold with no other input active. Input neurons were activated via single stimuli or in short bursts that generated 5 action potentials with an average frequency of 50 Hz in these neurons. The key parameter measured was the generation of an action potential in the output neuron indicating that transmission had occurred.

To determine whether the slow synaptic depolarizations increase the ability of fast EPSPs to excite action potentials in the output neurons, the slow EPSP producing input was excited and then the fast EPSP input was activated during the initial depolarizing phase of the slow depolarization (Fig 2B) or after 3 s during the sustained depolarization (Fig 2C). Initially subthreshold fast EPSPs triggered action potentials during the rising phase of the slow synaptic depolarization, but not during the sustained depolarization when their amplitudes were depressed. Even when the amplitudes of the fast and slow EPSPs were set so they were suprathreshold at rest (Fig 2D and 2E), they were still unable to evoke output neuron action potentials during large sustained synaptic depolarizations (Fig 2E).

We then used the model to investigate whether postsynaptic actions of endogenously released GABA can alter the ability of fast EPSPs triggered simultaneously in the recording neuron (ie the conduction delays in the input neurons were equal) to evoke action potentials in this neuron. The kinetics of GABA$_A$ channels produce depolarizations similar to fast EPSPs (Fig 1I) with the principal difference being that the reversal potentials for the two channels are -35 mV and 0 mV, respectively. Simultaneous activation of a 5 pulse (50 Hz) burst of subthreshold fast EPSPs and an identical burst of GABA$_A$ mediated depolarizations led to the output neuron firing 5 action potentials indicating that the GABA$_A$ depolarizations increase the probability of synaptic transmission (Fig 3C) in the network shown in Fig 3A. In contrast, when identical trains (with equal conduction delays) of sub-threshold fast EPSPs and GABA$_A$ synaptic depolarizations were delivered during the sustained phase of large slow synaptic depolarizations (network in Fig 3B), no action potentials were evoked (Fig 3D). The individual responses were much smaller than those seen in the absence of the slow depolarization (Fig 3C). This suppression of the combined fast EPSP and GABA$_A$ mediated responses depended on the magnitude of the slow synaptic depolarization, because the smaller slow depolarizations evoked by single stimuli did not alter firing produced by the combined fast responses with 5 pulse trains evoking 5 action potentials (Fig 3E).

Myenteric neurons express both GABA$_A$ and GABA$_C$ receptors [30], which differ dramatically in their kinetics [33]. As GABA released from nerve terminals might activate both receptor subtypes simultaneously, we examined how any synaptic potentials they produce may interact. Identical 5 pulse trains delivered simultaneously via GABA$_A$ and GABA$_C$ receptors evoked 2 action potentials followed by 3 fast depolarizations superimposed on a prolonged

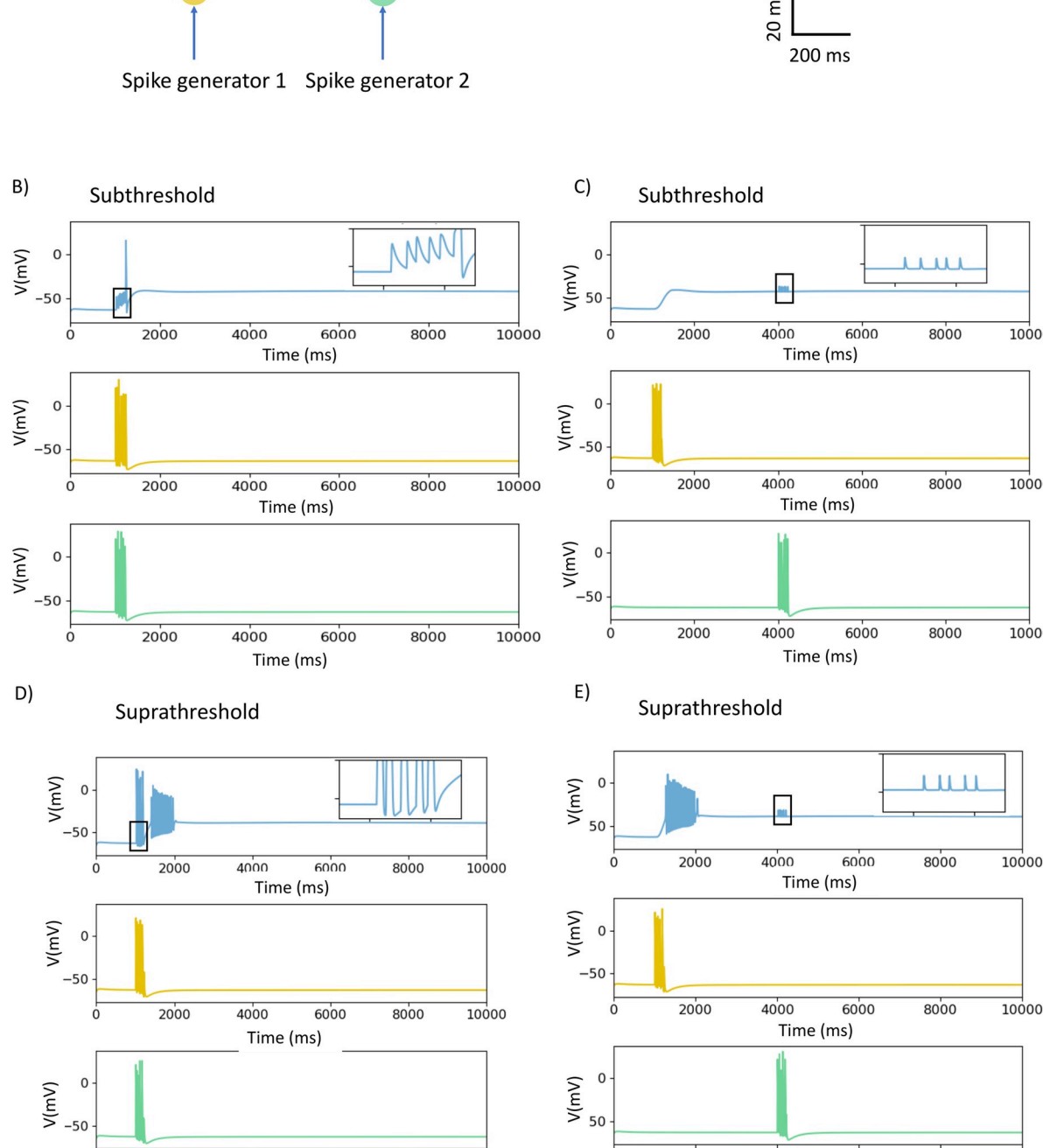

**Fig 2. Effects of interactions between fast EPSPs and slow synaptic depolarizations of differing amplitudes on firing of the recording neuron in a network model.** A) A schematic diagram of the structure of the network. The input neurons are connected to the recording neuron through slow synaptic depolarizations (indicated by sEPSP) and fast EPSPs (indicated by fEPSP) with synaptic weights set to produce responses that were either subthreshold (B, C) or suprathreshold (D, E) for triggering action potentials on their own. The input neurons are stimulated by spike generators. B) The responses of the neurons, membrane potential *v* (mV), to a train of 5 impulses, when the synaptic

weights are subthreshold. The fast EPSPs are activated during the initial depolarization of a subthreshold slow EPSP. The colour of the response of the neurons matches to the colour of the indicated neurons in A e.g., blue represent the response of the recording neuron. C) The responses of these neurons when subthreshold fast EPSPs are activated during the sustained depolarization of a slow EPSP. D) The responses of the neurons when synaptic weights were suprathreshold at rest and fast EPSPs were activated during the rising phase of the slow synaptic depolarization. E) The responses of the neurons when the synaptic weights were suprathreshold at rest. The fast EPSPs are activated during the sustained component of the slow synaptic depolarization. Boxes show regions of the membrane potential traces expanded in the inserts to allow comparisons of membrane potential changes during stimulus trains.

depolarization (network Fig 4A, result 4C). Thus, activation of $GABA_C$ receptors initially enhanced the ability of $GABA_A$ mediated depolarizations to trigger action potentials, but this did not persist throughout the stimulus train despite there being a depolarization that was sustained well past the end of the stimulus train.

The inability of the output neuron to fire during the sustained depolarization produced by activation of $GABA_C$ receptors was underlined when trains of 5 fast EPSPs and $GABA_C$-mediated depolarizations were delivered simultaneously to the recording neuron (network Fig 4B). When the amplitudes of both responses were set to being subthreshold, then the combined stimuli evoked 3 action potentials (Fig 4D) indicating that $GABA_C$ activation could facilitate firing evoked by a fast EPSP, but that this was not a one-to-one relationship. When the amplitudes of the fast EPSPs and of the $GABA_C$ responses were set to being suprathreshold, the combined stimuli evoked 5 spikes, but the later spikes were about 20 mV smaller due to shunting by the sustained chloride conductance increase produced by activating $GABA_C$ receptors (Fig 4E and 4F).

In Fig 2, we demonstrated that the sustained component of suprathreshold slow synaptic depolarizations suppressed firing evoked by suprathreshold fast EPSPs (Fig 2E) and that subthreshold slow depolarizations do not sum with subthreshold fast EPSPs to evoke action potentials (Fig 2C). To test if small slow synaptic depolarizations can inhibit firing evoked by summing initially subthreshold fast EPSPs and GABA evoked responses, we tested the effect of small slow depolarizations evoked by a single spike in the input neurons on synchronous subthreshold fast EPSPs and $GABA_C$ responses. The effect of summing these three inputs was to reduce the intermediate level of firing seen in the absence of the slow depolarization (compare 3 spikes in Fig 4D and 1 in Fig 5B). This indicates that even small slow synaptic depolarizations can depress firing over much of their time course.

## Interactions of fast and slow synaptic depolarizations–Ca imaging

$Ca^{2+}$ imaging, an indirect high throughput method of analysing neuronal activity, was used to test the prediction from the simulations that slow synaptic depolarization can inhibit the triggering of action potentials by fast EPSPs. This approach used mice that express the genetically encoded calcium indicator GCaMP6f in all neural crest derived cells (enteric neurons and glia), but not in other cell types of the GI tract. As shown in Fig 1, intracellular recordings show that a single electrical stimulus applied to a nerve trunk entering a myenteric ganglion evokes a fast EPSP (Fig 1E) in 70–90% of neurons [11,37,42,43], while a train of stimuli evokes a burst of fast EPSPs and often slow synaptic depolarizations in many neurons [11,37] (Fig 1F). We used these observations to set up an experimental protocol in which the $Ca^{2+}$ transients evoked by single stimuli on their own were compared with similar stimuli applied during the period after a train of stimuli that would have evoked an ongoing slow synaptic depolarization.

Overall, 13 myenteric ganglia from five animals were first electrically stimulated with a train of 20 pulses (20 Hz), to produce slow EPSPs in myenteric neurons in the mouse colon (Fig 1 [11]). In all, 464 neurons responded with $[Ca^{2+}]_i$ transients.

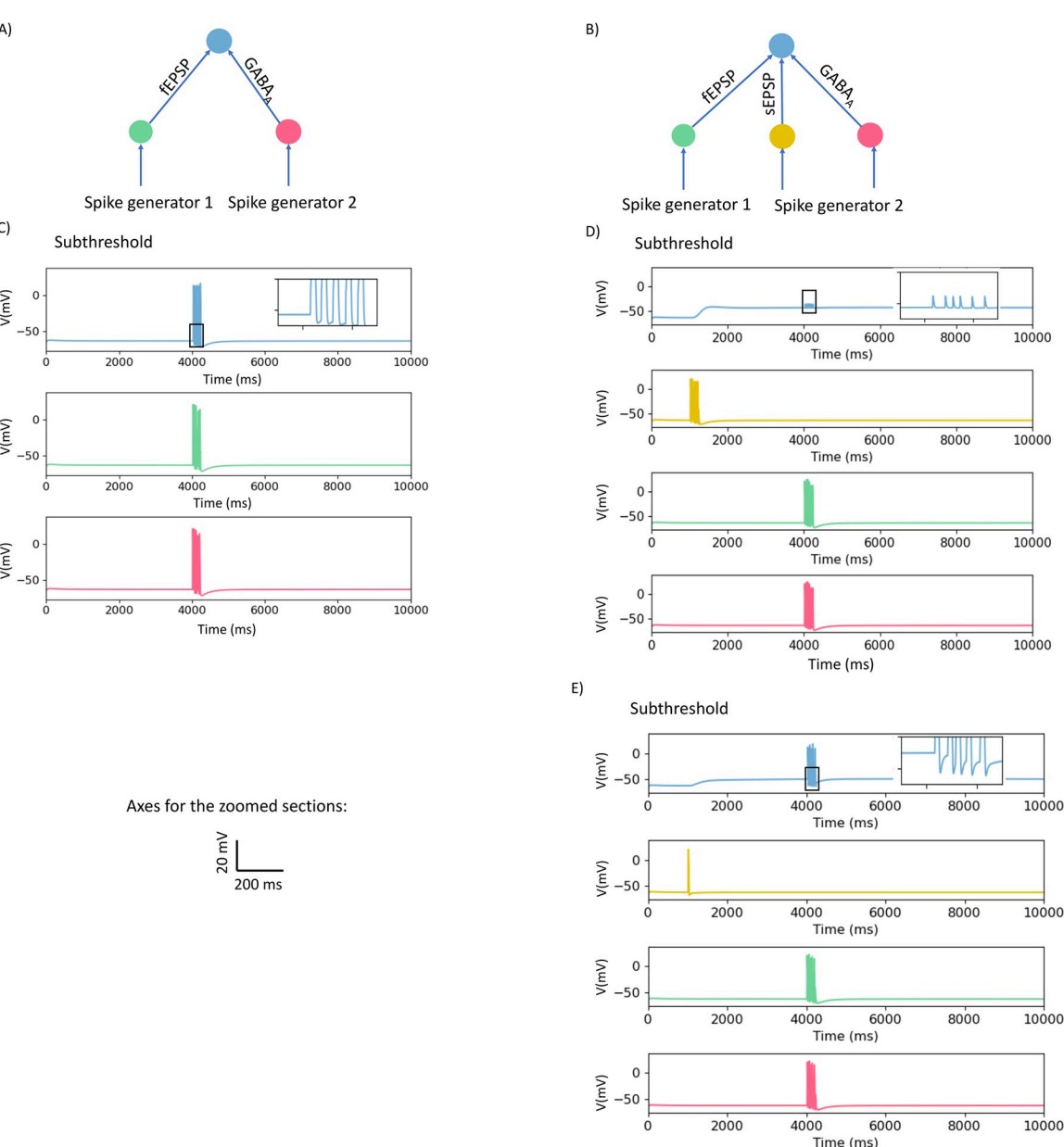

**Fig 3. Interactions between fast EPSPs, GABA$_A$ mediated synaptic responses and slow synaptic depolarizations on action potential firing in a model network.** A, B) Schematic diagrams of the structure of the networks. In A neurons are connected through fast EPSP (indicated by fEPSP), and GABA$_A$ synapses. In B neurons are connected through fast EPSPs, slow synaptic depolarizations (indicated by sEPSP), and GABA$_A$ synapses. C) Recording neuron responses when subthreshold fast EPSPs are activated synchronously with subthreshold GABA$_A$ synapses. The colour of the response of the neurons matches to the colour of the indicated neurons in A e.g., blue represent the response of the output neuron. D) The responses of the neurons when subthreshold fast EPSPs are activated simultaneously with subthreshold GABA$_A$ synapses during a subthreshold sustained depolarization evoked by a train of stimuli, no action potentials were evoked. E) The responses of the neurons when subthreshold fast EPSPs are activated simultaneously with subthreshold GABA$_A$ synapses during a small sustained depolarization evoked by a single stimulus, each fast EPSP evoked an action potential. Boxes show regions of traces shown in expanded form in the relevant inserts.

After 10 min, the ganglia were stimulated with a single pulse (1P) to evoke fast EPSPs in the great majority of myenteric neurons (Fig 1, [37]). This interval was chosen, because slow synaptic depolarizations and the $[Ca^{2+}]_I$ transients they evoke return to baseline within 1–5

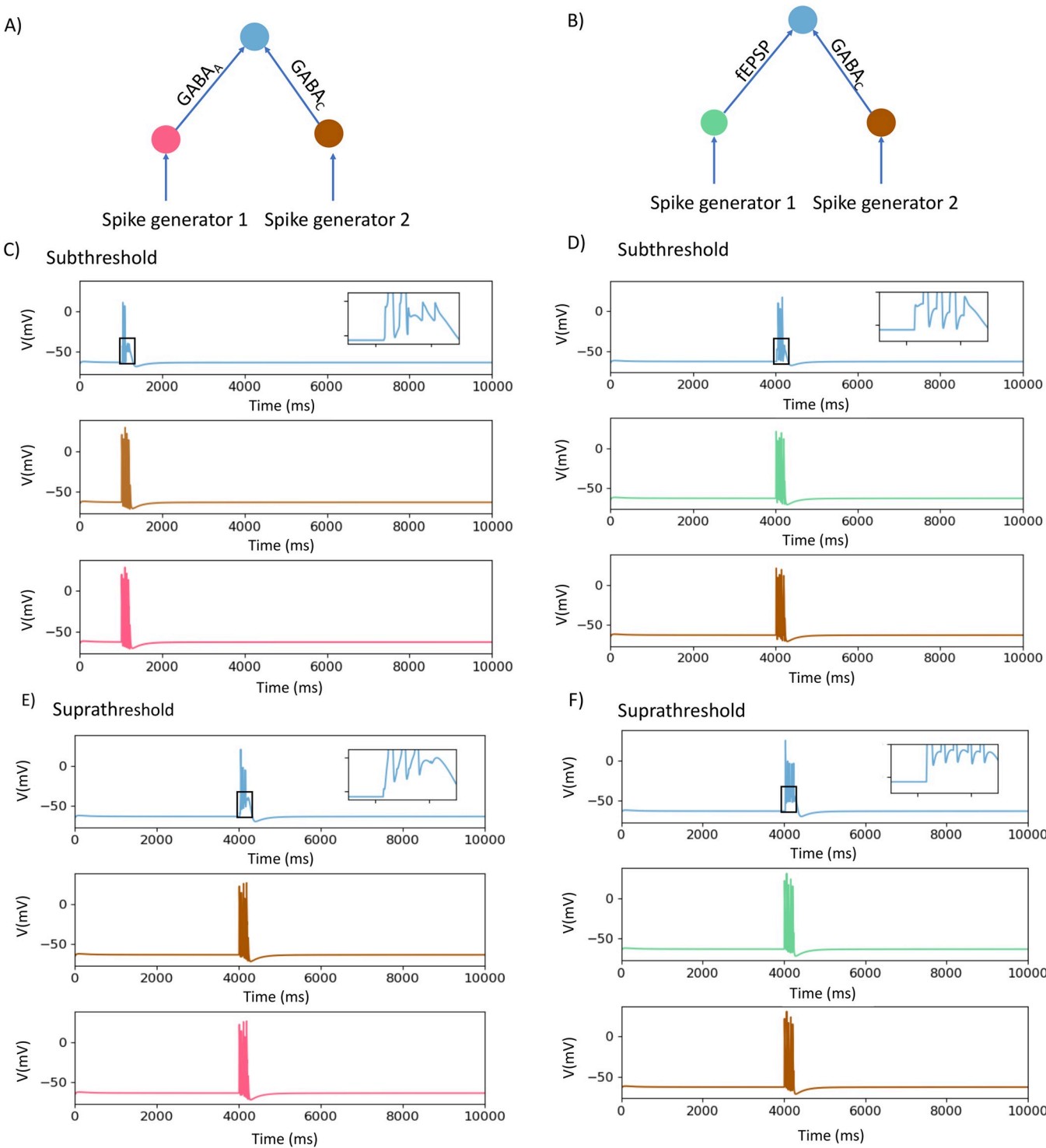

**Fig 4. Effects of synchronous activation of GABA$_C$ and GABA$_A$ or fast EPSP synapses on action potential firing.** A, B) Schematic diagrams of the networks. In A the neurons are connected through GABA$_A$ receptors and GABA$_C$ receptors. In B neurons are connected through fast EPSP GABA$_C$ synapses. C) when identical 5 pulse trains are delivered simultaneously with identical conduction times via GABA$_A$ and GABA$_C$ receptors with the synaptic weights set to being subthreshold. D) The responses when subthreshold fast EPSPs are activated in the together with subthreshold GABA$_C$ synapses. E) Responses of network 4A when GABA$_A$ and GABA$_C$ responses were set to being suprathreshold, 3 of 5 stimuli produced action potentials with the 2nd and 3rd spikes being truncated. F) Responses of network 4B when fast EPSPs and GABA$_C$ potentials were suprathreshold, all stimuli evoked spikes with later ones having smaller amplitudes. Boxes identify regions shown in expanded form in insets.

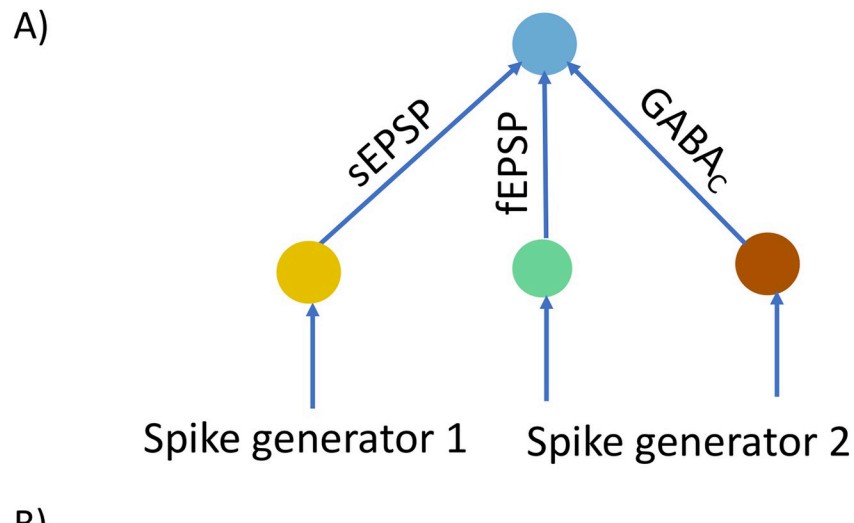

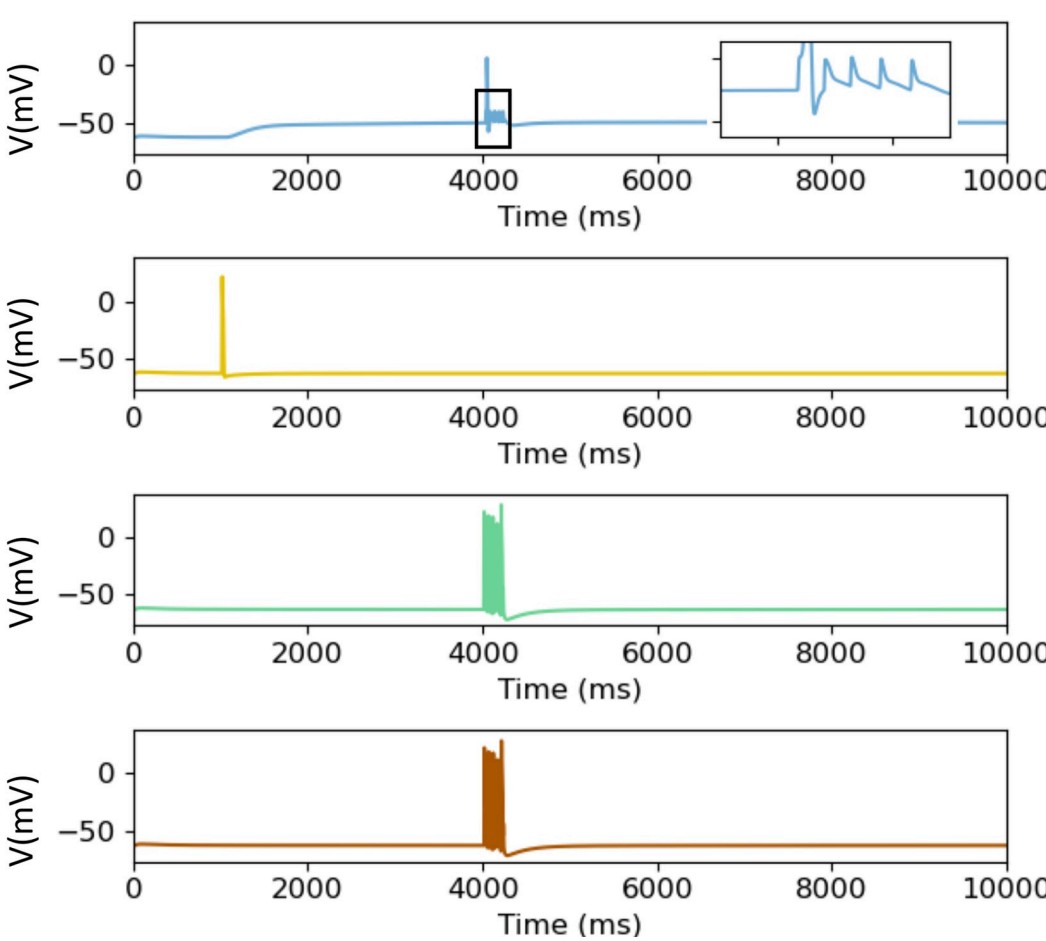

**Fig 5. Small slow synaptic depolarizations reduce firing induced by summation of subthreshold fast EPSPs and GABA_C mediated depolarizations.** The circuit is shown in panel A. B shows the response of the recording neuron to a 1 pulse stimulus delivered first to the slow depolarization neuron whose synaptic weight was set to the same value as that in Fig 3E and then after 3 s a 5 pulse stimulus to the fast EPSP and GABA_C neurons. The small slow synaptic depolarization depressed firing produced by the fast responses (compare to Fig 4D). The small box indicates the region of the trace shown in expanded form in the inset.

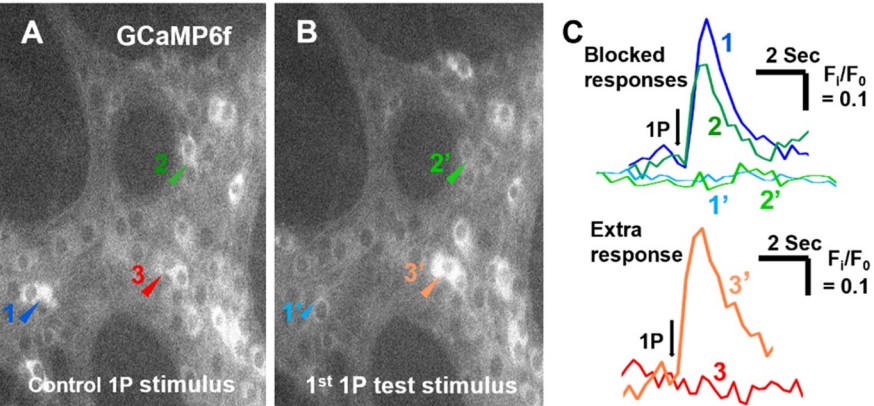

**Fig 6. Slow synaptic depolarizations suppress Ca$^{2+}$ transients evoked by stimuli that produce fast EPSPs in some myenteric neurons but increase Ca$^{2+}$ transients in others.** The control 1P stimulus (**A**) and the 1$^{st}$ 1P test stimulus (**B**) which were separated by a 20 P conditioning stimulus. Arrow heads indicate two neurons (1 blue, 2 green) that responded to the control (A) but not to the test (B) and one (3 red) that became responsive after the conditioning stimulus. The labels are colour-matched to their traces in (**C**).

minutes of the stimulus (see Fig 1 of this paper and Fig 3 of [30]), so the longer interval ensured that there was no summation of control and conditioning stimuli. Of the initial 464 responsive neurons, a total of 161 neurons exhibited a Ca$^{2+}$ transient after this 1P stimulus. After another 10 min interval, the same ganglia were stimulated with a second 20 pulse 20 Hz train and then 10 sec later with a 1P test stimulus (test), 78 of the 161 neurons that responded to the control 1P stimulus did not respond to this test 1P stimulus (Fig 6). About half (40) of those whose response was inhibited after 10 s, responded to a 2$^{nd}$ 1P test stimulus applied after another 10 s interval (20 s after the conditioning 20 P stimulus) indicating that the reduced excitability in these neurons was temporary and reversible.

The 20 P conditioning stimulus not only inhibited transients evoked by the test stimuli, but also excited test 1 transients in neurons that did not respond to the control stimulus. The 1$^{st}$ 1P trial evoked transients in 59 neurons that were initially unresponsive (Fig 6) and 30 of these plus another 27 initially unresponsive neurons exhibited transients evoked by the 2$^{nd}$ 1P test stimulation. It should be noted though that many myenteric neurons do not exhibit slow EPSPs in response to trains of stimuli, in such neurons the slow depolarization induced inhibition would be absent and other factors like release of other mediators that act presynaptically may increase firing evoked by fast EPSPs.

## Discussion

Synaptic interactions are the key to information processing in any network of neurons whether in the brain or the enteric nervous system. The key question is whether any particular combination of synaptic inputs to a specific neuron triggers an action potential or not. In this study, we investigated how 4 different types of synaptic input to individual enteric neurons can produce action potentials in these neurons using both computational simulation and Ca$^{2+}$ imaging to identify how summation of different synaptic inputs affects the output of a target neuron. The results indicate that depolarization of individual neurons via current injection from an intracellular recording electrode typically triggers 1 or a short high frequency burst of action potentials at the start of the depolarization, but this firing is transient (Fig 1). The compartmental model that we developed could mimic this firing pattern as long as the suite of ion channels in the cell membrane included a slowly activating voltage-dependent K channel, in

this study we included the Kv7.2 (*Kcnq2*) channel [3,39]. The primary means of transmission between enteric neurons is via fast EPSPs, but many enteric neurons also exhibit slow synaptic depolarizations especially in response to high frequency trains of action potentials in presynaptic axons (Fig 1; [8,22,37]. As expected, the model predicts that fast EPSPs occurring during the rising phase of slow synaptic depolarizations are more likely to trigger post-synaptic action potentials (Fig 2B and 2D). Paradoxically, the prolonged component of the slow synaptic depolarization inhibited action potential firing evoked by fast EPSPs despite the sustained depolarization being very close to the membrane potential required to trigger action potentials at rest (Fig 2E). This inhibitory effect of what are normally termed slow EPSPs was confirmed in the $Ca^{2+}$ imaging study, which showed that responses to stimuli that normally evoke fast EPSPs are inhibited in many myenteric neurons by stimuli that would evoke slow synaptic depolarizations.

We also simulated the effect of GABA release within the ENS because there is substantial indirect evidence that GABA has important synaptic effects on enteric neurons [28–30] despite the absence of direct evidence for GABA mediated synaptic potentials in enteric neurons. Both $GABA_A$ and $GABA_C$ receptors are expressed by enteric neurons [30] and the model shows that their activation would produce depolarizations with distinctly different time courses (Fig 2). The simulations indicate that $GABA_A$ receptor mediated synaptic potentials would be very similar to fast EPSPs and sum with them to trigger action potentials. On the other hand, the significantly slower $GABA_C$ mediated synaptic potentials have more complex effects with the consequences of their activation being biphasic: excitation followed by effective inhibition of action potential firing evoked by fast EPSPs.

## Role of Kv7.2

The basic model of individual neurons used in this study was adapted from a previously published compartmental model of an enteric neuron subtype, the intrinsic sensory neuron or AH neuron [38]. Intrinsic sensory neurons express several ion channels apparently absent from motor neurons and interneurons and have distinctively different firing properties, notably they exhibit a prolonged and substantial (up to 15 mV) hyperpolarization after their action potentials, hence the designation AH [7,42]. To simulate the behaviour of motor neurons and interneurons, which are electrophysiologically indistinguishable [7], we adopted the same strategy as that of our realistic compartmental model of the intrinsic sensory neuron with a suite of model ion channels tailored to the motor and interneuron firing patterns. This produced a model neuron that fired action potentials without a change in frequency throughout a 500 ms depolarizing current pulse (tonic firing) (Fig 1C), but both motor neurons and interneurons in mouse ENS usually only fire (at up to 150 Hz) within the first 75 ms of such a depolarization (Fig 1A). Inclusion of the M-current potassium channel Kv7.2 [39] converted the tonic firing to transient firing (Fig 1B) that mimicked the physiological behaviour of these neurons.

The relevant features of the Kv7.2 channel are that it is voltage-dependent, opens slowly compared with the voltage-dependent K channel responsible for repolarization during an action potential and inactivates very slowly or not at all when the triggering depolarization is maintained [39]. These features are shared with many other types of voltage-dependent K channels, so the Kv7.2 channel used in this study is representative of many different possible channels that could serve the same function. Nevertheless, *Kcnq2* is expressed by at least 2 enteric neuronal subtypes identified by single neuron RNAseq [3] making it a plausible candidate for producing transient firing in at least some myenteric neurons of the mouse. Further plausible candidates include other members of the KCNQ family of voltage dependent

channels (like Kv7.3 and Kv7.5) which are expressed by distinct populations of mouse enteric neurons [3,44].

## Slow synaptic depolarizations can be EPSPs or IPSPs within the one synaptic event

The most striking finding of this study is that prolonged depolarizations can suppress the generation of action potentials by fast EPSPs, even though the initial depolarization is excitatory. This conclusion is supported by both the computational simulations and indirect measurements of neuronal activity using $Ca^{2+}$ imaging. In each case, fast EPSPs activated several seconds after the initiation of a slow synaptic depolarization were unable to excite action potentials, when the same stimulus would trigger action potentials either in the absence of a slow depolarization or during the initial phase of such a depolarization.

The fast EPSPs in the computational study were modelled as a generic increase in inward current with a reversal potential of 0 mV and represent the sum of inputs being activated simultaneously. Most of these inputs use acetylcholine (ACh) acting via nicotinic ACh receptors (nAChR), but there is evidence for fast EPSPs mediated by ATP acting via P2X receptors and 5-HT acting via 5-HT$_3$ receptors [8,11,45–48], each of which have similar conductances and kinetics to the nAChR. The subthreshold fast EPSPs in the simulations can be assumed to occur physiologically when too few fast EPSP inputs are activated simultaneously to evoke an action potential, while the suprathreshold fast EPSPs in the simulations would occur when the stimulus recruits more presynaptic axons.

The computational simulations in this study did not include mechanisms by which activation of metabotropic receptors that mediate slow synaptic depolarizations could modulate the properties of the ligand-gated channels mediating fast EPSPs. Thus, the suppression of fast EPSP evoked firing during these depolarizations can only occur as a result of depolarization induced changes in the voltage-dependent ion channels incorporated in the model. Several such changes would be expected with the most notable being activation of the slowly inactivating Kv7.2 channel, which limits firing during depolarizations resulting from intracellular current injection, and delayed de-inactivation of the voltage-dependent sodium channels that carry the inward current of the action potentials. The latter might be responsible for the reduction in action potential amplitude that was seen in the bursts of action potentials triggered during the initial phases of synaptic depolarizations. But the constant frequency of firing during a prolonged depolarizing current step in the absence of Kv7.2 suggests that inactivation of Na$^+$ channels by ongoing depolarization may not have a major role in limiting firing produced by fast EPSPs.

The facilitation of fast EPSP induced firing by the rising phase of the slow synaptic depolarizations and the observation that when the latter are large enough they can trigger bursts of action potentials, both in the model and in intracellular recordings, justify the long standing identification of these events as slow EPSPs. Nevertheless, over the vast majority of their time course of 10 to 150 s the slow depolarizations act to suppress firing triggered by even suprathreshold fast EPSPs. Thus, they can be reasonably described as slow biphasic excitatory/inhibitory synaptic potentials despite being depolarizing throughout their time course. This conclusion stands regardless of the messenger systems that mediate the slow synaptic depolarization, which produce their membrane potential changes via changes in membrane conductance. There may be effects of the messengers on neuronal excitability, but these are not incorporated in the models we used, so such effects would be on top of the changes that we observed and should be examined using a model focusing on intracellular signalling.

It should be noted here that this applies to the slow depolarizations in interneurons and motor neurons. The intrinsic sensory neurons express a different suite of membrane ion channels and the messenger systems that underly slow synaptic depolarizations in these neurons directly modify the properties of some of these channels.

While the $Ca^{2+}$ imaging study confirmed that responses to trains of stimuli can produce a long-lasting inhibition of some neurons, it also indicates that other neurons exhibit a prolonged increase in excitability following such stimulus trains. The mechanisms responsible for this are not immediately obvious from the modelling, so although our model makes robust predictions and produces outputs that mimic the results of physiological experiments more needs to be incorporated to fully account for synaptic interactions in the ENS. One notable issue is that the RNAseq data indicate that the interneurons and motor neurons express heterogeneous populations of ion channels [3,44] implying the effects of prolonged depolarization probably differ between neuronal subtypes in ways not readily detected in intracellular recordings. Another is that presynaptic modulation of synaptic function is well established for enteric synapses (for a very recent example see [49]), although its functional significance is not well studied. Furthermore, the neurotransmitters and receptors responsible for slow synaptic depolarization are not yet fully identified [20] and their effects may not simply be on membrane potential, but may also be on the properties of the ligand gated channels that mediate fast EPSPs. For example, vasoactive intestinal peptide (VIP) can alter the properties of nAChR in parasympathetic ganglia [50] and VIP is found in synaptic terminals throughout the enteric nervous system [4]. Similarly, the tachykinin substance P is widely distributed within enteric nerve terminals [4] and can inhibit nicotinic ACh receptor channels [51]. Explaining the above mentioned discrepancy between model prediction and the $Ca^{2+}$ imaging study will require dissection of synaptic functions to a much greater level of detail than currently available.

## GABA can excite enteric neurons, but has a biphasic effect when it activates GABA$_C$ receptors

The equilibrium potential for Cl$^-$ in enteric neurons is probably 15–25 mV positive to resting membrane potential [41] and hence more positive than the threshold for triggering action potentials. This is why activation of GABA$_A$ receptors excites action potentials if enough GABA is released and lesser amounts can produce depolarizations that sum with fast EPSPs to trigger firing. Activation of GABA$_C$ receptors can excite spikes, but because of the slower kinetics of GABA$_C$ channels also produces a slow depolarization to a membrane potential close to E$_{Cl}$ and summation of this later depolarization with either fast EPSPs or GABA$_A$ mediated depolarizations has complex effects. When either fast EPSPs or GABA$_A$ mediated responses are subthreshold, they can sum with the initial depolarization produced by GABA$_C$ activation to initiate spikes, but this effect is not sustained during short trains of stimuli. Rather the early synaptic potentials fire spikes, but the later synaptic events in a 5 pulse train fail to do so. This is presumably because the sustained depolarizations produced by GABA$_C$ receptors activate the same membrane dependent changes that inhibit firing during slow synaptic depolarizations. Another mechanism involved in this inhibition is likely to be the prolonged opening of chloride channels leading to shunting of inward currents and suppression of the regenerative increase in Na conductance that underlies neuronal action potentials, which is reflected by the smaller amplitudes of action potentials that are triggered during this late depolarization. The late inhibition phase of the GABA$_C$ response in the model may account for our previously published finding that blocking GABA$_C$ receptors enhances $Ca^{2+}$ transients evoked in myenteric neurons by trains of electrical stimuli [30].

## Physiological implications

The results of this study go some way to providing explanations for two of the enduring mysteries of enteric neuroscience. First, how to account for the prolonged time frames of many of the motor behaviours of the GI tract given the rapid signalling within the neural circuits? For example, fast EPSPs last 20–50 ms, action potentials in enteric axons conduct at 20–50 cm/s and neurons fire in short bursts lasting up to 100 ms, but colonic contractile complexes in mouse repeat at 2–5 minute intervals, can have durations of 10–30 s and propagate at around 0.3 cm/s [52–54]. Thus, major motor behaviours are several orders of magnitude slower than the activity at the level of synaptic interactions. It has long been postulated that slow EPSPs in intrinsic sensory neurons form one of the bridges between these differing time scales, with the prolonged after-hyperpolarizing potentials in these neurons also being involved [55,56]. However, the finding that slow synaptic depolarizations in interneurons and motor neurons can be initially excitatory and then become inhibitory adds a new perspective to this problem as it may account for the intervals between contraction complexes. Further understanding will come from future models of the circuits together with identification of the mechanisms underpinning the slow synaptic depolarizations in interneurons and motor neurons to allow their manipulation in future physiological experiments.

The second mystery has been the relative absence of inhibitory synaptic transmission in the ENS when compared with the central nervous system. Some inhibitory synaptic potentials have been identified in myenteric neurons [10,22,27], but these are typically small and most neurons lack such inputs. That synaptic inhibition is needed in enteric circuits is indicated by the anatomical and electrophysiological data showing that there are several recurrent excitatory networks within the ENS including one formed by intrinsic sensory neurons [57,58] and at least one other formed by interneurons within different motor pathways [59]. Such networks require activity dependent inhibition to prevent uncontrolled firing in the elements of the network [57]. Firing in the intrinsic sensory neuron network appears to be regulated by the AHPs that are characteristic of these neurons [57,60], but the source of inhibition of the interneuron networks has not been determined. Several theories have been proposed including presynaptic inhibition, retrograde actions of nitric oxide and occasional inhibitory synaptic potentials, but all are unsatisfactory. The demonstration that what has been thought to be an excitatory synaptic potential can actually also be inhibitory for prolonged periods adds yet another possibility. This is attractive because slow synaptic depolarizations are more common than inhibitory synaptic potentials in myenteric neurons. Further, slow synaptic depolarizations produce an intrinsic pattern within the firing of the interneurons whose consequences will need to be explored in future computational simulations.

The present study treated all synaptic events produced by activation of a neuron as having a constant amplitude and each neuron as being representative of the entire population of neurons producing a particular type of synaptic potential. In reality, several different neurons producing fast EPSPs would be expected to converge on a single target neuron and the amplitudes of the fast EPSPs they evoke would vary both between neurons and between successive action potentials. This type of "noise" and related randomness like channel openings, variations in synaptic delays and spontaneous synaptic potentials due to physiological activity in realistic networks would be expected to have some effect at the micro-level on the probability of firing of the target neuron. The circuit building software that we used to construct our test networks is capable of generating circuits incorporating these and other elements. Nevertheless, we chose to exclude these random elements to test the specific questions of macroscopic interactions of defined synaptic potentials. Analysis of how random variations affect firing in the network is a much larger task and well beyond the scope of this study.

Metabotropic receptors responding to classical and peptide neurotransmitters with changes in intracellular messengers that can modify membrane channels to produce slow depolarizations are, however, widespread within the central and peripheral nervous systems. These receptors are often termed neuromodulatory and in this role they are thought to alter transmission by altering the properties of neurotransmitter receptors. The results of this study suggest that the effects of slow changes in membrane potential produced by activation of such metabotropic receptors on the properties of voltage-dependent ion channels should also be considered to be part of their ability to modify neurotransmission and information processing.

## Materials and methods

### Ethics statement

All experimental procedures on mice were approved the University of Melbourne Animal Experimentation Ethics Committee (approval number 1714308).

### Computational modelling

We developed a conductance-based model of the S neurons (interneurons and motor neurons) in the GIT using the NEURON simulation environment. Each neuron simulated was modelled as: soma 10 segments, diameter 25 μm, length 49 μm, membrane capacitance 1 μF.cm$^{-2}$. This corresponds to a large myenteric neuron in guinea-pig and represented a compromise between human and small rodent neurons. Preliminary studies varying the different dimensions by a factor of 2 showed that there was no qualitative change in the behaviour of the neurons when their size was altered within normal anatomical ranges. The ion channel properties of the neurons are modelled based on a Hodgkin-Huxley formalization. The rising phase of the action potential is due to the inward current through sodium (Na$^+$) channels. There are two types of Na$^+$ channels in the model: Na$_V$1.3 and Na$_V$1.7 channels.

**Na$_V$1.3 channel.** Most of the inward current is provided by Na$_v$1.3 channels ($I_{Na_v1.3}$), which has one activation gate and slow and fast inactivation gates. The Na$_v$1.3 current is described as

$$I_{Na_v1.3} = g_{Na_v1.3} m_1^3 m_2 m_3 (v - E_{Na}),  \quad (1.1)$$

where $g_{Na_v1.3}$ is the maximum conductance, $m_1$ defines the variable for the activation gate, $m_2$ is the fast inactivation gate, $m_3$ is the slow inactivation gate, $v$ is the membrane potential and $E_{Na}$ the is the reversal potential for Na$^+$ [38]. The values of the different parameters in the model are gathered in Table 1. The details of the kinetics of the activation and inactivation gates ($m_1$, $m_2$, and $m_3$) are described as

$$m_{1\alpha} = \frac{0.4(v + 31)}{1 - \exp\left(\frac{-v-31}{4.5}\right)},  \quad (1.2)$$

$$m_{1\beta} = \frac{0.124(-30 - v)}{1 - \exp\left(\frac{v+31}{4.5}\right)},  \quad (1.3)$$

$$m_{1\tau} = \frac{q_{Na_V1.3}}{m_{1\alpha} + m_{1\beta}},  \quad (1.4)$$

$$m_{1\infty} = \frac{m_{1\alpha}}{m_{1\alpha} + m_{1\beta}},  \quad (1.5)$$

**Table 1. The parameters in the model and their associated values, the references listed show the papers from which the specific values were taken.**

| Name | Parameter | Value | References |
|---|---|---|---|
| maximum conductance (Na$_v$1.3) | $g_{Na_v1.3}$ | 0.01 | [38] |
| reverse potential | $E_{Na}$ | 55 mV | [38] |
| maximum conductance (Na$_v$1.7) | $g_{Na_v1.7}$ | 0.01 | [38] |
| maximum conductance (delayed rectifier) | $g_{K_{dr}}$ | 0.01 | [38] |
| reverse potential for potassium | $E_k$ | -85 mV | [38] |
| maximum conductance (A-type potassium) | $g_{k_A}$ | 0.0014 | [38] |
| A-type potassium channel's kinetics constant | $m_\tau$ | 0.5 | [38] |
| A-type potassium channel's kinetics constant | $h_\tau$ | 15 | [38] |
| maximum conductance (Kv7.2) | $g_{kv7.2}$ | 0.15 | [62] |
| time constant (Kv7.2) | $\tau_{kv7.2}$ | 20.7 ms | [62] |
| half activation potential | $V_h$ | 49.8 | [62] |
| slope factor (Kv7.2) | $k_1$ | 37.26 | [62] |
| slope factor (Kv7.2) | $k_2$ | 118.5 | [62] |
| reverse potential (fast EPSP) | $E$ | 0 mV | – |
| decay time constant (fast EPSP) | $\tau_2^\alpha$ | 5ms | – |
| rising time constant (fast EPSP) | $\tau_1^\alpha$ | 1ms | – |
| reverse potential (chloride) | $E_{cl}$ | -35 mV | [40, 41] |
| decay time constant (GABA$_A$) | $\tau_2^A$ | 5.6 ms | – |
| rising time constant (GABA$_A$) | $\tau_1^A$ | 0.285 ms | – |
| decay time constant (GABA$_C$) | $\tau_2^C$ | 50 ms | [33] |
| rising time constant (GABA$_C$) | $\tau_1^C$ | 20 ms | – |
| production rate of cAMP (sEPSP) | $\alpha_1$ | 0.22 | [63] |
| cAMP removal rate (sEPSP) | $\beta_1$ | 0.41 | [63] |
| production rate of catalytic subunit (sEPSP) | $\alpha_2$ | 0.22 | [63] |
| catalytic subunit removal rate (sEPSP) | $\beta_2$ | 0.27 | [63] |
| phosphorylation rate (sEPSP) | $\alpha_3$ | 0.22 | [63] |
| rate of returning to the unphosphorylated state (sEPSP) | $\beta_3$ | 0.12 | [63] |

where $q_{Na_V1.3}$ is the temperature sensitivity for the sodium current and is defined as

$$q_{NaV_{1.3}} = 2^{\frac{T-24}{10}}, \tag{1.6}$$

and $T$ is the temperature in Celsius.

Fast inactivation of the sodium current is described as

$$m_{2\alpha} = \frac{0.03(-45-v)}{1 - exp\left(\frac{v+45}{1.5}\right)}, \tag{1.7}$$

$$m_{2\beta} = \frac{0.01(-45-v)}{1 - exp\left(\frac{v+45}{1.5}\right)}, \tag{1.8}$$

$$m_{2\tau} = \frac{q_{Na_V1.3}}{m_{2\alpha} + m_{2\beta}}, \tag{1.9}$$

$$m_{2\infty} = \frac{1}{1 + exp\left(\frac{v+50}{4}\right)}. \tag{1.10}$$

Slow inactivation of the gate is described as

$$m_{3\alpha} = exp\left(\frac{1156.8(v+60)}{8.315(273.16+T)}\right), \tag{1.11}$$

$$m_{3\beta} = exp\left(\frac{231.36(v+60)}{8.315(273.16+T)}\right), \tag{1.12}$$

$$m_{3\tau} = \frac{m_{3\beta}}{0.0003(1+m_{3\alpha})}, \tag{1.13}$$

$$m_{3\infty} = \frac{1}{1 + exp\left(\frac{v+58}{2}\right)} + M\left(1 - \frac{1}{1 + exp\left(\frac{v+58}{2}\right)}\right), \tag{1.14}$$

where $M$ is a constant and its value determines the maximum percentage of the channel that can enter the slow inactivated state.

**Na$_v$1.7 channel.** The other sodium channel is Na$_v$1.7, which has a similar structure to the Na$_v$1.3 channel with different values for the parameters, and results in a small inward sodium current [38]. The current is described by

$$I_{Na_v1.7} = g_{Na_V1.7}m_1^3 m_2 m_3 (v - E_{Na}), \tag{2.1}$$

where $g_{Na_v1.7}$ is the maximum conductance rate, $m_1$ is the activation gate, $m_2$ is the fast inactivation gate, $m_3$ is the slow inactivation gate. The activation of the channel is described by

$$m_{1\alpha} = \frac{15.5}{1 + exp\left(\frac{v-5}{-12.08}\right)}, \tag{2.2}$$

$$m_{1\beta} = \frac{35.2}{1 + exp\left(\frac{v+72.7}{16.7}\right)}, \tag{2.3}$$

$$m_{1\tau} = \frac{1}{m_{1\alpha} + m_{1\beta}}, \tag{2.4}$$

$$m_{1\infty} = \frac{m_{1\alpha}}{m_{1\alpha} + m_{1\beta}}. \tag{2.5}$$

Fast inactivation of the gate is defined as

$$m_{2\alpha} = \frac{0.38685}{1 + exp\left(\frac{v+122.35}{15.29}\right)}, \tag{2.6}$$

$$m_{2\beta} = \frac{2}{1 + exp\left(\frac{v+5.53}{12.7}\right)} - 0.003, \tag{2.7}$$

$$m_{2\tau} = \frac{1}{m_{2\alpha} + m_{2\beta}}, \tag{2.8}$$

$$m_{2\infty} = \frac{m_{2\alpha}}{m_{2\alpha} + m_{2\beta}}. \tag{2.9}$$

The slow inactivation of the channel is described as

$$m_{3\alpha} = 0.00003 + \frac{0.00092}{1 + exp\left(\frac{v+93.9}{16.6}\right)}, \tag{2.10}$$

$$m_{2\beta} = 132.05 - \frac{0.00092}{1 + exp\left(\frac{v+93.9}{16.6}\right)}, \tag{2.11}$$

$$m_{2\tau} = \frac{1}{m_{3\alpha} + m_{3\beta}}, \tag{2.12}$$

$$m_{3\infty} = \frac{m_{3\alpha}}{m_{3\alpha} + m_{3\beta}}. \tag{2.13}$$

The falling phase of the action potential is due to three potassium channels: an A-type potassium current, a delayed rectifier current, and the current of the $KV_{7.2}$ channel.

**Delayed rectifier channel.** The delayed rectifier current provides a significant amount of the outward current for the model, which results in the downstroke of the action potential. This current is modelled as

$$I_{K_{dr}} = g_{K_{dr}} m(v - E_k), \tag{3.1}$$

where $g_{K_{dr}}$ is the maximum conductance rate and $E_k$ is the reversal potential for the potassium current with the values shown in Table 1. The channel $m$ is described by

$$m_\infty = \frac{1}{1 + \exp\left(\frac{-v-25}{5}\right)}, \tag{3.2}$$

$$m_\tau = 0.27 \frac{0.5}{1 + \exp\left(\frac{v+27}{156}\right)}. \tag{3.3}$$

**A-type potassium channel.** The A-type potassium current also contributes to the falling phase of the action potential but it has a very small amplitude compared to the delayed-rectifier current and is described as

$$I_{k_A} = g_{k_A} m_1^3 m_2(v - E_k), \tag{4.1}$$

where $g_{k_A}$ is the maximum conductance rate of the current, $E_k$ is the potassium reversal potential, $m_1$ is the activation of the channel, and $m_2$ is the inactivation of the channel described by

$$m_{1\infty} = \frac{1}{1 + \exp\left(\frac{-v-30}{8}\right)}, \tag{4.2}$$

$$m_{2\infty} = \frac{1}{1 + \exp\left(\frac{v+80}{6}\right)}. \tag{4.3}$$

The values of $m_\tau$ and $h_\tau$ are constant for A-type potassium current and these values are gathered in Table 1 [38].

**Kv$_{7.2}$ channel.** The other potassium channel, named as Kv$_{7.2}$, tends to suppress the firing of neurons. Kv$_{7.2}$ is a principal subunit of the slow voltage-gated M-channels [39,61]. The main role of these channels is to control the excitability of the neurons; they prevent the repetitive firing and convert tonic firing to phasic firing. M-channels activate at sub-threshold voltages. They have a very slow inactivation, which stabilizes the membrane potential and contributes to the resting membrane potential. The reason for the dampening effect of M-channels is that the spike discharge increases the membrane conductance, which results in an increase of the outward current and, consequently, the spike burst increases the threshold for the subsequent spikes. Blockade of this channel results in a change in the threshold of the action potential and facilitates repetitive spike discharges. The Kv$_{7.2}$ channel is modelled based on the Hodgkin-Huxley framework as

$$I_k = g_{kv7.2} {m_1}^3 m_2 (v - E_k), \tag{5.1}$$

where $g_{kv7.2}$ is the maximum conductance, $E_k$ is the reversal potential for potassium, $m_1$ is the fast activation gate variable, and $m_2$ is a slow activation gate variable.

The fast activation gate has the first-order kinetics illustrated in Fig 7. The dynamics of the activation gate are expressed by

$$\dot{m}_1 = \alpha(1 - m_1) - \beta m_1. \tag{5.2}$$

The solution to the first order of the differential equation is

$$m_1 = m_{1\infty}(1 - \exp(-t/\tau_{kv7.2})), \tag{5.3}$$

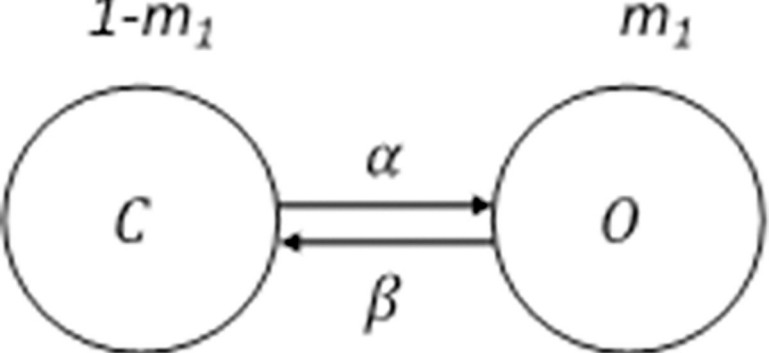

**Fig 7. The kinetics of the activation gate of the Kv$_{7.2}$ channel.** The activation gate has first order kinetics. O refers to the open state and C represents the closed state. $m_1$ is the state variable. $\alpha$ and $\beta$ are the forward and backward rate variables.

where the steady-state and time constant ($m_{1\infty}$ and $\tau_{kv7.2}$) of the state variable are voltage dependent. Since

$$\frac{dm_1}{dt} = \frac{m_{1\infty} - m_1}{\tau}, \tag{5.4}$$

by substituting Eq 5.2 into Eq 5.4, we will have

$$\frac{m_{1\infty} - m_1}{\tau} = (\alpha + \beta)\left(\frac{\alpha}{\alpha + \beta} - m_1\right), \tag{5.5}$$

in which,

$$\tau = \frac{1}{\alpha + \beta}, \tag{5.6}$$

$$m_{1\infty} = \frac{\alpha}{\alpha + \beta}. \tag{5.7}$$

The forward and backward transition variable, of $\alpha$ and $\beta$, are described as

$$\alpha = \frac{\exp\left(\frac{v - v_h}{k_1}\right)}{18.5 + 2.17\left(\exp\left(\frac{v - v_h}{k_1}\right) + \exp\left(-\frac{v - v_h}{k_1}\right)\right)}, \tag{5.8}$$

$$\beta = \frac{\exp\left(-\frac{v - v_h}{k_1}\right)}{18.5 + 2.17\left(\exp\left(\frac{v - v_h}{k_1}\right) + \exp\left(-\frac{v - v_h}{k_1}\right)\right)}, \tag{5.9}$$

where $v_h = 49.8$ is the half activation potential and $k_1 = 37.26$ is the slope factor [62].The slow activation gate also has a first order kinetics with different parameters of the transition rate compared to the fast activation gate,

$$\alpha = \frac{\exp\left(\frac{v - v_h}{k_2}\right)}{154.75 + 15.62\left(\exp\left(\frac{v - v_h}{k_2}\right) + \exp\left(-\frac{v - v_h}{k_2}\right)\right)}, \tag{5.10}$$

$$\beta = \frac{\exp\left(-\frac{v - v_h}{k_2}\right)}{154.75 + 15.62\left(\exp\left(\frac{v - v_h}{k_2}\right) + \exp\left(-\frac{v - v_h}{k_2}\right)\right)}, \tag{5.11}$$

where $k_2 = 118.5$ is the slope factor. The slow activation gate has a longer time constant than the activation gate; therefore, it is around 8 times slower than the activation gate [62].

## Fast synaptic depolarization

The fast EPSP synapses are modeled based on two-exponential kinetics with two different time constants for the rising phase and the decay phase of the current as

$$i = g(v - E), \tag{6.1}$$

$$g = \exp(-t/\tau_2^\alpha) - \exp(-t/\tau_1^\alpha), \tag{6.2}$$

where $E$ is the reversal potential and $\tau_2^\alpha$ and $\tau_1^\alpha$ are time constants of different phases of the

gating kinetics of the receptors. The reversal potential is 0mV and the time constants are $\tau_2^\alpha = 5$ ms and $\tau_1^\alpha = 1$ ms.

## Modelling ionotropic GABA receptors

The model of the gating kinetics of $GABA_A$ and $GABA_C$ receptors can be divided into three main phases: activation, deactivation, and desensitization. The activation phase refers to the rising phase of the synaptic current, which depends on two factors: opening rate after the transmitter binds to the receptor and the amount of transmitter present. There is a delay in the raising phase of the current. The synaptic current is modelled as

$$i = g(v - E_{cl}),\tag{7.1}$$

$$g = \exp(-t/\tau_2^R) - \exp(-t/\tau_1^R),\tag{7.2}$$

where $E_{cl}$ is the reversal potential and $\tau_2^R$ and $\tau_1^R$ are time constants of different phases of the gating kinetics of the receptors. R indicates the type of the receptors, either $GABA_A$ or $GABA_C$. The reversal potential for the chloride channel is –35 mV [41]). The time constants for the $GABA_A$ receptors are $\tau_2^A = 5.6$ ms and $\tau_1^A = 0.285$ ms and for the $GABA_C$ synapses are $\tau_2^C = 50$ ms and $\tau_1^C = 20$ ms.

## Slow synaptic depolarizations

Slow synaptic depolarizations are modelled based on activation, inactivation, and summation of cAMP and protein kinase A-dependent second messenger cascades [60]. The first stage of the slow EPSP, which relates to the production and removal of cAMP, can be described with the kinetic scheme,

$$\dot{D} = \alpha_1 I(t) - \beta_1 D,\tag{8.1}$$

where $D$ is the amount of cAMP, $I(t)$ is the amount of agonist, $\alpha_1$ is the rate of the production of cAMP, and $\beta_1$ is the cAMP removal rate from the system [63]. The second stage of the model describes the production of cAMP though the activation of PKA. PKA is composed of two regulatory subunits and two catalytic subunits. The kinetics of this stage is described by the second order differential equation,

$$\dot{C} = \alpha_2 D^2 - \beta_2 C,\tag{8.2}$$

where $C$ is the amount of catalytic subunit, $\alpha_2$ is the rate of the production of $C$ and $\beta_2$ is the rate constant for the removal of $C$ from the system [63]. The final stage, which describes the phosphorylation of a potassium channel, is modelled using the following differential equation,

$$\dot{P} = -\alpha_3 CP + \beta_3(1 - P),\tag{8.3}$$

where $P$ is the amount of unphosphorylated channel, $\alpha_3$ is the rate of phosphorylation, and $\beta_3$ is the rate of returning to the unphosphorylated state [63].

## Model network

The model of the network was developed using the NEURON simulation environment and the NetPyNE tool [64]. The neurons in the network are connected through fast EPSPs and slow synaptic depolarizations, and $GABA_C$ and $GABA_A$ synaptic potentials. Fig 4B shows a schematic diagram of one network. The neurons are connected to the recording neuron through fast EPSP, slow EPSP and $GABA_A$ synapses. The neurons in the model are

stimulated by either a single spike or a series of spikes initiated by two spike generators with different ranges of time intervals between the spikes. The time intervals between the spikes of the spike generator randomly vary between 20–30 ms. The duration of the train of spikes is 200 ms. The input to the neuron connected to the recording neuron by slow synaptic depolarizations is provided by the first spike generator and the inputs to the neurons connected to the recording neurons by fast EPSPs and GABA$_A$ synapses are provided by the second spike generator.

## Calcium imaging experiments

**Tissue preparation.**   *Wnt1-Cre;R26R-GCaMP6f* mice of either sex aged 8–12 weeks in which neural crest-derived cells, including enteric neurons and glia, express the genetically encoded calcium indicator, GCaMP6f, were used. *Wnt1-Cre;R26R-GCaMP6f* mice were bred by mating heterozygous *Wnt1-Cre* males with heterozygous floxed GCaMP(B6J.Cg-Gt(ROSA) 26Sor$^{tm95.1(CAG-GCaMP6f)Hze}$/MwarJ) females in standard mouse boxes in the biological research facility. Pups were kept with their mothers until weaning and then housed in sex specific boxes containing 4–6 mice until they were old enough for experiments. Both parent strains were on a C57BL6 background.

Mice were killed by cervical dislocation and a segment of proximal colon was removed from each mouse and quickly placed in physiological saline (composition in mM: NaCl 118, NaHCO$_3$ 25, D-glucose 11, KCl 4.8, CaCl$_2$ 2.5, MgSO$_4$ 1.2, NaH$_2$PO$_4$ 1.0) bubbled with carbogen gas (95% O$_2$, 5% CO$_2$). The colonic segments were then cut along the mesenteric border, stretched, and pinned flat mucosal side up in a Petri dish lined with a silicone elastomer (Sylgard 184; Dow Corning, North Ryde, NSW, Australia).

The mucosa and submucosa were removed using ultra-fine microdissection forceps, then the tissues were turned over and the longitudinal muscle layer was stripped away to obtain preparations of myenteric plexus attached to the circular muscle layer (CMMP). The CMMP preparations were immobilized by stretching the tissue (plexus uppermost) over an inox ring which was then clamped by a matched rubber O-ring [65]. A maximum of two rings were prepared from each segment of proximal colon. The tissue was transferred to a recording bath for imaging, where it was superfused (1 ml/min) with 95% O2: 5% CO2 equilibrated physiological saline at room temperature throughout the experiment via a gravity-fed inflow system.

**Imaging.**   Ringed preparations were imaged with a resolution of 512 x 512 pixels using a Plan-Apochromat 20x/1,0 DIC (UV) VIS-IR M27 water dipping objective, with a numerical aperture of 1 and a 1x software zoom on an upright Zeiss (Axio Examiner Z.1) microscope. Using a (Axiocam 702) camera (Carl Zeiss Microscopy, North Ryde, NSW, Australia), images (16 bit) were acquired at 7 Hz.

Myenteric ganglia were electrically stimulated with a single pulse and a train of pulses (20 pulses, 20 Hz; Master-8 pulse stimulator [A.M.P.I, Israel], connected to a stimulation isolation unit ISO-Flex, [A.M.P.I, Israel]) using a focal stimulating electrode (tungsten wire; 50 μm) placed on an inter-ganglionic fibre tract entering the chosen ganglion.

**Protocols.**   *Interactions of slow depolarizations and fast EPSPs* To test the effects of slow synaptic depolarizations on neuronal excitability, the electrical stimulation regime included delivery of a train of pulses (20 pulses 20 Hz) to an internodal strand and then 10 minutes later a single pulse was delivered to the same internodal strand (1P control). After a further 10 mins, a 20 pulse train of pulses was delivered, and the single pulse stimulus was repeated after 10 sec (test 1) and again after another 10 sec interval (test 2).

### Data analysis and statistics

Analyses of the $Ca^{2+}$ imaging data were performed using custom-written directives in IGOR Pro (WaveMetrics, Lake Oswego, Oregon, USA) [66,67]. Regions of interest were drawn over a selected area of the cytoplasm for each neuron that responded to electrical stimulation. A neuronal response was measured by the change in fluorescence of each neuron, where the change in amplitude of the intracellular calcium ($[Ca^{2+}]_i$) transient signal for each response from baseline amplitude is calculated ($\Delta F_i/F_0$). $[Ca^{2+}]_i$ transients were only considered if the intensity of the transient signal was more than 10 times the intrinsic noise. The number of responding neurons from the 1P control stimulus were counted and their $\Delta F_i/F_0$ analysed and the same neurons were re-examined to determine how many responded to the test 1 and test 2 stimuli, i.e., after the conditioning 20 pulse stimulus.

$\Delta F_i/F_0$ for each responsive neuron was compared between control and test stimuli were compared using unpaired t-tests with $P < 0.05$ considered statistically significant, but no significant differences between responses were identified. Comparisons were performed using GraphPad Prism 5.0 (GraphPad Software, San Diego California). However, over 50% of the control responders failed to respond to test 1 so it was felt that it was not meaningful to pool both responsive and unresponsive test 1 neurons to compare mean $\Delta F_i/F_0$ to the controls. Accordingly, data are presented in the text as number of responding neurons with recovery of responses at test 2 indicating that any inhibition was likely to be real.

## Author Contributions

**Conceptualization:** Parvin Zarei Eskikand, Joel C. Bornstein.

**Data curation:** Rachel M. Gwynne.

**Formal analysis:** Parvin Zarei Eskikand, Katerina Koussoulas.

**Funding acquisition:** Joel C. Bornstein.

**Investigation:** Parvin Zarei Eskikand, Katerina Koussoulas, Rachel M. Gwynne.

**Methodology:** Parvin Zarei Eskikand, Katerina Koussoulas, Rachel M. Gwynne.

**Project administration:** Joel C. Bornstein.

**Resources:** Joel C. Bornstein.

**Software:** Parvin Zarei Eskikand.

**Supervision:** Joel C. Bornstein.

**Validation:** Parvin Zarei Eskikand.

**Writing – original draft:** Parvin Zarei Eskikand.

**Writing – review & editing:** Katerina Koussoulas, Rachel M. Gwynne, Joel C. Bornstein.

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
