## [Decision Letter · Decision Letter 0]

11 Jan 2022

Dear Dr. Bornstein,

Thank you very much for submitting your manuscript "Computational simulations and Ca2+ imaging reveal that slow synaptic depolarizations (slow EPSPs) inhibit fast EPSP evoked action potentials for most of their time course in enteric neurons" for consideration at PLOS Computational Biology. As with all papers reviewed by the journal, your manuscript was reviewed by members of the editorial board and by several independent reviewers. The reviewers appreciated the attention to an important topic. Based on the reviews, we are likely to accept this manuscript for publication, providing that you modify the manuscript according to the review recommendations.

Sincerely,

Joanna Jędrzejewska-Szmek, Ph.D.

Associate Editor

PLOS Computational Biology

Daniele Marinazzo

Deputy Editor

PLOS Computational Biology

[LINK]

Reviewer's Responses to Questions

**Comments to the Authors:**

Reviewer #1: Yes, uploaded as attachment

Reviewer #2: This manuscript examines the interaction between fast and slow postsynaptic potentials (PSPs) in enteric neurons by combining computational modeling with Ca+2 imaging. The work addresses how such interactions can drive firing of action potentials. After presenting their simulations, the authors describe exploratory experiments in which imaging of a genetically encoded Ca+2 sensor (GCaMP6f) is used to assess how different types of synaptic interactions can result in excitation and inhibition.

The investigation is interesting, addresses an important issue and is generally well-done.

Enteric neurons form a complex network within the gastrointestinal tract that is essential for normal digestive function and whose dysregulation contributes to major disease states. The network contains different functionally specialized neuronal cell types that express a variety of neurotransmitters, receptors, and ion channels. Fast excitatory PSPs are mediated by ligand gated ion channels that respond to acetylcholine, serotonin and ATP. These different receptors gate cationic conductances with similar kinetics and a reversal potential near 0 mV. GABA also mediates fast PSPs through anionic chloride conductances with a reversal potential near -35mV. Because this reversal potential is depolarized to threshold for action potential generation, such responses are also excitatory. Two types of GABA receptor – A and C – have been identified in enteric neurons and have different kinetics. GABA–A responses are fast, similar to cationic fast EPSPs while GABA–C responses are considerably slower. Slow EPSPs are metabotropic in nature and mediated in this system by small molecule transmitters such acetylcholine, and by peptidergic modulators such as substance P. Although the ionic basis of slow EPSPs is diverse, many of them are mediated by the suppression of voltage-gated K+ conductances formed by KCNQ (Kv7) channels.

The authors used a conductance based neuronal model constructed with NEURON software to analyze the interaction between fast cationic EPSPs, fast anionic PSPs, and slow EPSPs. Many of the results are as expected. Summation of subthreshold fast EPSPs and fast anionic PSPs can drive action potentials. A novel finding arises when suprathreshold fast EPSPs are timed to coincide with steady depolarization during a slow EPSP. Then it becomes evident that the slow EPSP can act to inhibit excitation (Fig. 2). The authors go on to show that action potentials generated by summation of subthreshold fast EPSPs and anionic PSPs can be inhibited if evoked during the steady plateau phase of a slow EPSP (Figs. 3,5). These finding are interesting because they are counter-intuitive, thereby demonstrating the usefulness of a computation approach. In the simulations, the authors vary the magnitude of slow EPSP using different numbers of presynaptic stimuli. Based on the computational predictions they then do experiments with living preparations of isolated enteric ganglia. Using electrical stimulation with different numbers of shocks, they monitor gCaMP signals in fields of cells and make inferences about whether the responses reflect interaction between fast and slow PSPs and whether they reflect excitation or possible inhibition as predicted by the simulations. The authors interpret these experiments as being consistent with predictions from the computational modeling.

Although the manuscript was very intriguing, there are several ways in which it could be strengthened.

1) In Figure 1, the authors argue that their model replicates physiological recordings and is therefore valid. While this appears reasonable, it is less clear how the authors chose parameters for all the conductances and their kinetics. For example, the peak Na+ and delayed rectifier conductances (Table 1) are said to have values of 0.01, but with no units! The Kv7.2 conductance as value of 0.15 (no units). Generally, the magnitude of m-conductance is much smaller than that of voltage-gated Na+ conductance and the delayed rectifier K+ conductance. The table should be carefully edited and the strategy for tuning the model by setting parameters should be explained more clearly. It would also help to add a citation for evidence that the reversal potential for chloride is -35mV. Also, what was the resting potential, input resistance and capacitance of the model? – these often are important in tuning a model.

2) In many of the figures, starting with fig 1, the lettering is way too small. Figure 1 would also be improved by denoting physiological recordings and simulations in the figure, not simply in the legend. It took a while to decode this and annotate the manuscript, which detracts from the story.

3) The slow EPSP is modeled based on the kinetics of cyclic AMP generation and downstream kinases. While this may have been reasonable when the model was originally developed many years ago, it is now clear the Kv7 suppression is mediated by the hydrolysis of PIP2. Perhaps it is sufficient to use a model that simply replicates the kinetics and stimulus dependence of slow EPSPs (analogous to how fast PSPs were modeled).

4) Does Na+ channel inactivation play a critical role in the inhibition seen during slow EPSPs? – see line 437 in discussion. This seems highly likely. The authors could address the question in 3 ways. One would be to monitor the state if inactivation during the computational simulation. The second would be to inject a hyperpolarizing current during the slow EPSP to prevent the depolarization. Third, they could map the time course for the ‘inhibition’ during the slow epsp by probing it with fast PSPs and different time points and determining whether inhibition correlates with the degree of depolarization or the duration of depolarization.

5) Statistics – the methods (lines 766 to 770) describe statistical tests using unpaired t-test. This ready may have overlooked it, but did not see any reports of averaged data, variance or statistical tests. Given this approach, it would be more appropriate to describe the experimental results as exploratory findings that possibly demonstrate a proof of principle for further work.

Reviewer #3: In this paper by Eskikand et al., a computational model is presented that allows the authors to predict the impact of fast and slow synaptic potentials and ionotropic Gaba receptor activation on postsynaptic neurons. Interestingly, they arrive at the conclusion that slow excitatory potentials may only be exciting at their onset but actually be inhibitory later on. Their robust modelling is corroborated by a Ca imaging approach that shows indeed that shortly after such a slow excitatory potential a single stimulus is not able to induce a Ca transient any longer. I enjoyed reading this paper and have only a number of minor questions and suggestions that might further improve the manuscript

It would be informative to also show the results in the output neuron of the different stimulations on an expanded timescale (e.g. in an inset), such that amplitude changes and possibly small delays can be appreciated.

The authors compare their model with actual electrophysiological recordings, showing a close match between what is recorded and what is generated in silico. I was intrigued by the hyperpolarizing pulses that were used during the e-phys recordings to estimate the input resistance. What happens if they were introduced as inputs into the model? Does the in silico version show the same output?

Figure 2: it is not clear why the panels are organized like that. To me it seems more logic to show the synchronous and a-synchronous stimulation panels respectively above one another.

The Ca imaging data are intriguing and important to provide support for the inhibitory role that an excitatory potential might have. However, the fact that some neurons start responding rather than being inhibited is a little bit detached from the rest of the paper, despite the fact that this is fairly well addressed in the discussion, with some options clearly explained as to why this might happen. I suggest to add a little bit of that explanation already in the results, to avoid that this section ends a bit on a confusing note, and only gets explained a couple of pages later.

The authors refer to the guinea-pig study of Cherubini and North to suggest that the reversal potential of Cl in colonic neurons is around -35 mV. Interestingly this was also shown in mouse colonic neurons in a patch clamp study by Kang et al. DOI: 10.1152/ajpcell.00437.2002. There it was suggested that these neurons accumulate Cl. Could this be another possible explanation for the differences that the authors observe in their Ca imaging results? What if not all neurons accumulate Cl to the same extent? Could this explain why some and not other neurons are inhibited during the prolonged duration of the sEPSP?

I suggest to present the Ca imaging data a little bit differently. Wouldn’t it be more logic to show the number of responses per ganglion rather than only mentioning total of un/responding neurons summed over all the experiments?

Furthermore, it is also important to show that the neurons were back to baseline before the 1P stimulus was given. Telling from the snapshots shown that is indeed the case, thus showing the values would strengthen the conclusions. Imagine that a Ca transient induced by the 20Hz stimulation lasts longer than 10 s, then the indicator would be still saturated and therefore a 1P stimulation would not be able to elicit a transient, not because there is no additional depolarization or AP but simply because the reporter is not available. As said, this does not seem to be the case as seen in the 2 images. However, the resting values (preferably not normalized) before the 20Hz and 1P stimulation would therefore be informative.

When referring to the role of nACh receptors in synaptic responses in the mouse ENS, it seems appropriate to also refer to the paper of Foong et al. 2015 (doi: 10.1523/JNEUROSCI.4175-14.2015).

Why is there only one temperature sensitive component in the model? It seems that only Nav1.3 is temperature sensitive? Is that correct? What effect did temperature have on the model? And does that match the Ca imaging results, which were recorded at RT ?

The legend of Fig. 3 claims that the colors of the neurons are matched, but – at least in the pdf version of the manuscript- there is quite a bit of difference between the pinkish colors. Please correct.

Introduction line 155: mv should be mV

Legend of Fig.4: … are activated in the… I think a word is missing here.

Line 391: although commonly used in the ENS research field, “AH” has not been defined in the current paper yet.

Line 411: I assume the verb is missing: … which are expressed

**Have the authors made all data and (if applicable) computational code underlying the findings in their manuscript fully available?**

Reviewer #1: Yes

Reviewer #2: **No: **code link is ok, data link Sharepoint to Dr. Bornstein is broken

Reviewer #3: Yes

PLOS authors have the option to publish the peer review history of their article (what does this mean?). If published, this will include your full peer review and any attached files.

Reviewer #1: **Yes: **Wolfgang Kunze

Reviewer #2: **Yes: **John P Horn, PhD

Reviewer #3: No

Figure Files:

Data Requirements:

Reproducibility:

References:

---

## [Decision Letter · Decision Letter 1]

3 May 2022

Dear Dr. Bornstein,

We are pleased to inform you that your manuscript 'Computational simulations and Ca2+ imaging reveal that slow synaptic depolarizations (slow EPSPs) inhibit fast EPSP evoked action potentials for most of their time course in enteric neurons' has been provisionally accepted for publication in PLOS Computational Biology.

Best regards,

Joanna Jędrzejewska-Szmek, Ph.D.

Associate Editor

PLOS Computational Biology

Daniele Marinazzo

Deputy Editor

PLOS Computational Biology

Reviewer's Responses to Questions

**Comments to the Authors:**

Reviewer #1: An outstanding paper and in view of the importance of the enteric nervous system to general health, life and gut to brain transmission, and additionally the role played by the enteric nervous system in gut to brain transmission in an era of computer modelling and artificial intelligence, this paper should be of general interest to the intelligent public.

Reviewer #2: The authors have made revisions that provide an adequate response to the issues raised in the initial round of reviews. NO further comment.

Reviewer #3: My questions and concerns have been addressed adequately. I am not completely sure though whether I agree with the statement "as there is no reason to believe that there would be systematic differences between ganglia", but this can be subject for future studies. Congratulations on the nice paper.

**Have the authors made all data and (if applicable) computational code underlying the findings in their manuscript fully available?**

Reviewer #1: Yes

Reviewer #2: **No: **link to modeling code is ok. Link to raw data a unimelbourne does not work.

Reviewer #3: Yes

PLOS authors have the option to publish the peer review history of their article (what does this mean?). If published, this will include your full peer review and any attached files.

Reviewer #1: **Yes: **Wolfgang Kunze

Reviewer #2: **Yes: **John P Horn

Reviewer #3: No

---

## [Editor Report · Acceptance letter]

30 May 2022

PCOMPBIOL-D-21-02193R1 

Computational simulations and Ca2+ imaging reveal that slow synaptic depolarizations (slow EPSPs) inhibit fast EPSP evoked action potentials for most of their time course in enteric neurons

Dear Dr Bornstein,

I am pleased to inform you that your manuscript has been formally accepted for publication in PLOS Computational Biology. Your manuscript is now with our production department and you will be notified of the publication date in due course.

With kind regards,

Zsofia Freund
